# Antiplatelet and Antithrombotic Effects of Isaridin E Isolated from the Marine-Derived Fungus via Downregulating the PI3K/Akt Signaling Pathway

**DOI:** 10.3390/md20010023

**Published:** 2021-12-24

**Authors:** Ni Pan, Zi-Cheng Li, Zhi-Hong Li, Sen-Hua Chen, Ming-Hua Jiang, Han-Yan Yang, Yao-Sheng Liu, Rui Hu, Yu-Wei Zeng, Le-Hui Dai, Lan Liu, Guan-Lei Wang

**Affiliations:** 1Department of Pharmacology, Zhongshan School of Medicine, Sun Yat-sen University, Guangzhou 510080, China; pann@mail2.sysu.edu.cn (N.P.); lzc3@outlook.com (Z.-C.L.); l1352272780@163.com (Z.-H.L.); yanghy58@mail2.sysu.edu.cn (H.-Y.Y.); liuysh55@mail2.sysu.edu.cn (Y.-S.L.); hurui8@mail2.sysu.edu.cn (R.H.); zengyw25@mail2.sysu.edu.cn (Y.-W.Z.); 2Institute of Pediatrics, Guangzhou Women and Children’s Medical Centre, Guangzhou Medical University, Guangzhou 510080, China; 3School of Marine Sciences, Sun Yat-sen University, Guangzhou 510080, China; chensenh@mail.sysu.edu.cn (S.-H.C.); jiangmh23@mail2.sysu.edu.cn (M.-H.J.); 4Department of Basic Medical Sciences, Zhongshan School of Medicine, Sun Yat-sen University, Guangzhou 510080, China; dailh3@mail2.sysu.edu.cn

**Keywords:** isaridin E, marine fungal natural products, platelets, aggregation, antithrombotic, PI3K

## Abstract

Isaridin E, a cyclodepsipeptide isolated from the marine-derived fungus *Amphichorda felina* (syn. *Beauveria felina*) SYSU-MS7908, has been demonstrated to possess anti-inflammatory and insecticidal activities. Here, we first found that isaridin E concentration-dependently inhibited ADP-induced platelet aggregation, activation, and secretion in vitro, but did not affect collagen- or thrombin-induced platelet aggregation. Furthermore, isaridin E dose-dependently reduced thrombosis formation in an FeCl_3_-induced mouse carotid model without increasing the bleeding time. Mechanistically, isaridin E significantly decreased the ADP-mediated phosphorylation of PI3K and Akt. In conclusion, these results suggest that isaridin E exerts potent antithrombotic effects in vivo without increasing the risk of bleeding, which may be due to its important role in inhibiting ADP-induced platelet activation, secretion and aggregation via the PI3K/Akt pathways.

## 1. Introduction

Arterial occlusive diseases, such as ischemic heart disease or stroke, are leading causes of mortality and morbidity worldwide [1,2]. Platelet activation is at the center of acute pathological processes in those patients. Additionally, it is triggered by the rupture or erosion of the atherosclerotic plaque, leading to the local activation of coagulation and release of adhesive molecules as well as various platelet agonists [3,4]. These agonists mainly include adenosine diphosphate (ADP), which binds to purinergic receptors (P2Y_1_ and P2Y_12_), thrombin, collagen, thromboxane A2, and 5-hoxytryptamine. The pharmacological blockage of surface receptors is a powerful antiplatelet therapy in the treatment and prevention of arterial thrombosis [5,6]. For example, stimulation of the P2Y_12_ receptor by ADP amplifies the activation response to other platelet activators and results in sustained aggregation and secretion; therefore, P2Y_12_ receptor inhibitors effectively reduce the ischemic risk in the management of cardiovascular diseases clinically [7]. However, all contemporary antiplatelet and antithrombotic drugs bear a major clinical limitation of bleeding and the risk of thrombosis may persist despite the presence of potent antithrombotic therapy [6,8]. Therefore, it is still an urgent need to develop novel antiplatelet drugs with improved efficacy and safety. 

Marine-derived natural compounds are excellent sources of novel bioactive compounds [9,10,11,12,13,14,15]. Recently, accumulating novel antithrombotic compounds from marine sources has attracted increased attention [16]. Isaridin E, a fungal cyclodepsipeptide, is the main secondary metabolite of the marine-derived fungus *Amphichorda felina* (syn. *Beauveria felina*) SYSU-MS7908. The fungi were collected from an ascidian *Styela plicata* of North Atoll of the Xisha Islands (17°06′14.50″ N and 111°28′35.03″ E) in the South China Sea [17]. In addition to its insecticidal activity [18], Fang-Rong Chang and colleagues found that isaridin E exerted anti-inflammatory activity in human neutrophils by inhibiting superoxide anion generation and elastase release induced by formyl-l-methionyl-l-leucyl-l-phenylalanine (FMLP) without cytotoxicity [19]. Oxidative stress leads to excessive platelet activation, which has been closely associated with platelet hyperactivity and hyperaggregability in cardiovascular diseases including stroke, atherothrombosis, and myocardial infarction [20,21,22,23,24]. In turn, activated platelets stimulated overproduction of ROS that further amplifies the platelet activation and aggregation signaling [21,25]. Considering its potent inhibitory effects on superoxide anion generation, we therefore hypothesized that isaridin E may regulate platelet activation and aggregation.

In our present study, we initially evaluated the effects of isaridin E on platelet functions including platelet aggregation, secretion and activation in vitro. We found that isaridin E concentration-dependently inhibited ADP-induced platelet aggregation, activation and secretion in vitro, but did not affect collagen- or thrombin-induced platelet aggregation. Thus, we further examined whether isaridin E affected arterial thrombus formation in vivo on an FeCl_3_-induced carotid artery thrombosis mouse model. Given the integrating role of PI3K/Akt signaling during the ADP-induced, platelet-mediated thrombosis formation, we also investigated the effects of isaridin E on the PI3K/Akt signaling pathway. 

## 2. Results

### 2.1. Chemical Structure of Isaridin E

Isaridin E was obtained as a colorless crystal with a molecular formula of C_35_H_54_O_7_N_5_, as identified in the HR-ESIMS spectrum (*m/z* 656.40183 [M + H]^+^ calcd for C_35_H_54_O_7_N_5_, 656.40178) (Appendix A). The ^1^H and ^13^C NMR data (Appendix A) of the molecule were pure and well in agreement with the reference [26]. Thus, the compound was confirmed as isaridin E, as cyclo-[(OH) (*S*)-HMPA^1^-l-Pro^2^-l-Phe^3^-N-Me-l-Val^4^-N-Me-l-Val^5^-*β*-Ala^6^(CO)]; the chemical structure of isaridin E is presented in Figure 1.

### 2.2. Isaridin E Had No Toxicity in Platelets and No Influence on General Hemostasis

We first examined whether isaridin E has cytotoxic effects on washed mouse circulating platelets using the LDH (lactate dehydrogenase) cytotoxicity assay kit. Destruction of the cell membrane caused by apoptosis or necrosis leads to the release of enzymes in the cytoplasm into the culture medium, including LDH, with stable enzyme activity. Therefore, the quantitative analysis of the percentage of LDH leakage from the platelets is commonly used to evaluate the cytotoxic effects of tested compounds, and LDH > 10% was considered cytotoxic. As shown in Table 1, isaridin E showed no obvious cytotoxicity on washed platelets at concentrations of 0, 6.25, 12.5, 25, 50, 100, 200, or 400 μM (*p* > 0.05). Next, we tested whether treatment with isaridin E for 3 h (a single intragastric administration at 100 mg/kg) affects hematological and coagulation parameters in vivo. As shown in Appendix A, no significant differences in analyzed hematological parameters, such as platelet counts, mean platelet volume, red blood cell counts, mean red blood cell volume, or white cell counts, were apparent between the normal saline (NS)-treated group, vehicle-treated group and isaridin E-treated group. Meanwhile, isaridin E showed no obvious effects on general coagulation in mice, as assessed by activated partial thromboplastin time (APTT), thrombin time (TT), prothrombin time (PT), and fibrinogen (FIB) (Appendix A). These results indicate that isaridin E has no obvious influence on the blood cell counts and the blood coagulation cascade.

The tail bleeding assay was used to further evaluate the hemostatic properties of isaridin E [27]. As shown in Figure 2, compared with the vehicle control, clopidogrel (10 mg/kg) significantly increased the tail bleeding time. However, treatment with isaridin E (12.5, 25, 50 or 100 mg/kg) showed no obvious effects on the bleeding time as compared with the vehicle control (Figure 2). There was also no statistically significant difference in the bleeding time among different concentrations of isaridin E (12.5, 25, 50, or 100 mg/kg, Figure 2). These findings were consistent with the effect of isaridin E on coagulation parameters, suggesting that isaridin E may affect platelet functions but not the blood coagulation cascade.

### 2.3. Isaridin E Inhibited ADP-Induced Platelet Aggregation and Release In Vitro

We initially investigated whether isaridin E affected mouse platelet aggregation in vitro using the light transmission aggregometry, a method used as the gold standard [28]. Clopidogrel is a commonly used platelet inhibitor and antithrombotic agent that targets the P2Y_12_ receptor. We thus set clopidogrel as the positive control in the current study. The result shown in Figure 3 summarizes the effects of isaridin E on the maximum platelet aggregation in vitro in response to thrombin (0.02 U/mL), collagen (1 μg/mL), and ADP (5 μM); there were no significant differences in the maximum platelet aggregation in response to thrombin and collagen among all the groups (Figure 3A,B, *p* > 0.05). As shown in Figure 3C, isaridin E at 12.5, 25, 50, 100, or 200 μM significantly inhibited the ADP-induced maximum platelet aggregation to 75.6 ± 2.6%, 55.7 ± 4.8%, 26.6 ± 3.5%, 15.6 ± 2.9%, and 8.1 ± 0.8% of vehicle control, respectively. In comparison with the vehicle control, isaridin E at 50 μM significantly reduced the ADP-induced platelet aggregation by 73.4 ± 3.5% at an extent similar to clopidogrel (20 μM). The IC_50_ value of isaridin E inhibiting ADP-induced platelet aggregation was 26.1 ± 2.9 μM. Moreover, isaridin E exerted a concentration-dependent inhibition of ADP-induced platelet ATP release (Figure 3D). It was found that 6.25 and 12.5 μM isaridin E could reduce ADP-induced platelet ATP release to 60.7 ± 7.9% and 31.7 ± 4.8% that of the vehicle control, respectively, suggesting its potent inhibitory effects on ADP-induced platelet ATP release. The IC_50_ value of isaridin E inhibiting ADP-induced platelet ATP release was 7.9 ± 1.1 μM.

### 2.4. Isaridin E inhibited Platelet Activation In Vitro

The activation of platelets is crucial for platelet function, and inhibiting platelet activation is the main treatment strategy for thrombotic diseases [29]. Isaridin E inhibited ADP-induced platelet aggregation and ATP release in a concentration-dependent manner; therefore, we further examined whether isaridin E affected platelet activation. Upon platelet activation, P-selectin (CD62p) is released from secretory granules and transferred to the surface of activated platelets. Thus, the surface expression of CD62p on platelets is commonly used as an important biomarker of platelet activation [30]. As shown in Figure 4, washed mouse platelets were pretreated with vehicle or isaridin E (25, 50, or 100 μM) for 30 min, followed by stimulation with 50 μM of ADP for 5 min. As shown in Figure 4, compared with the negative control, 50 μM of ADP markedly increased the percentage of CD62p expression on the platelet surface from 2.6 ± 0.9% to 35.6 ± 1.7%. Isaridin E decreased ADP-induced CD62p surface expression in a concentration-dependent manner, from 35.6 ± 1.7% to 29.6 ± 1.0% (25 μM), 16.0 ± 1.4% (50 μM), and 8.4 ± 1.0% (100 μM). 

### 2.5. Isaridin E Inhibited Fecl_3_-Induced Carotid Artery Thrombus Formation In Vivo 

The above in vitro results (Figure 3 and Figure 4) indicated that isaridin E has antiplatelet effects; thus, we investigated whether isaridin E exhibited antithrombotic activity in vivo by using the FeCl_3_-induced carotid artery thrombosis mouse model. Multiple intragastric administrations of vehicle or isaridin E (12.5, 25, 50, or 100 mg/kg) were given at 1, 24, and 48 h before the onset of FeCl_3_ injury. Clopidogrel (10 mg/kg), a well-established antithrombotic drug in clinical therapy, was set as a positive control. As shown in Figure 5, isaridin E significantly prolonged the time to the first thrombus formation and reduced the maximum thrombus size in a dose-dependent manner (Figure 5B,C), indicating that isaridin E could dose-dependently alleviate thrombus formation and vessel occlusion after FeCl_3_ injury to the mouse carotid artery. Moreover, the antithrombotic effects of isaridin E at 50 or 100 mg/kg were comparable to that of clopidogrel at 10 mg/kg.

### 2.6. Effects of Isaridin E on the PI3K/Akt Signaling Pathway In Vitro

The above results revealed that isaridin E may exert the antithrombotic effects through regulating ADP-induced platelet activation and aggregation signaling pathways. The activation of PI3K and its downstream effector, Akt, is a common signal pathway downstream of several platelet receptors (e.g., P2Y_12_ and PAR4), which is critical in modulating platelet hyperactivity during ischemic cardiovascular diseases [31]. Therefore, PI3K/Akt signaling is an attractive target for the development of new antiplatelet drugs [29]. Activation of PI3K and Akt occurs as a result of phosphorylation upon stimulation with various platelet agonists including ADP [32]; therefore, we investigated the influence of isaridin E on the ADP-induced phosphorylation of PI3K and Akt. As shown in Figure 6, ADP (5 μM) significantly upregulated the phosphorylation of PI3K and Akt in human megakaryocytic cell line (Meg-01 cells); however, isaridin E at 50 μM significantly suppressed the enhanced phosphorylation of PI3K and Akt induced by ADP without altering the total protein expressions of PI3K and Akt.

## 3. Discussion

In the current study, we observed inhibitory effects of isaridin E on ADP-induced platelet activation, secretion, and aggregation in vitro, as well as antithrombotic properties on an FeCl_3_-induced carotid artery thrombosis mouse model. We also found that isaridin E appears to be quite safe because isaridin E at 400 μM does not induce toxic effects on mouse platelets. To the best of our knowledge, we demonstrated, for the first time, that isaridin E is a safe marine metabolite which displays strong antiplatelet and antithrombotic activities without significant influences on bleeding time, platelet counts, and coagulation parameters. Moreover, we found that isaridin E significantly suppressed the ADP-induced phosphorylation of PI3K and Akt, suggesting that the PI3K/Akt signaling pathway may be a potential action target of isaridin E. 

Our results provide the first direct evidence that isaridin E concentration-dependently inhibits ADP-induced mouse platelet activation, a key event in the development of platelet hyperactivity and arterial thrombosis during cardiovascular diseases with increased thrombotic risk [33,34]. ADP is an endogenous agonist, causing aggregation which is crucial for physiological hemostasis and pathological thrombosis [35]. Upon platelet activation, ADP can also be secreted from dense granules by numerous agonists and is responsible for secondary aggregation regardless of the initial agonists. Therefore, the ADP purinoceptor P2Y_12_ is an important therapeutic target for antiplatelet drugs. Thienopyridines and other ADP-receptor antagonists, including clopidogrel, cangrelor and ticagrelor, are highly successful in reducing ischemic cardiovascular events via the inhibition of ADP receptor P2Y_12_. We found that isaridin E specifically reduced ADP-induced platelet aggregation, ATP release, and platelet activation in a concentration-dependent manner, although it exhibited no effects on collagen- or thrombin-induced aggregation, suggesting that the specific inhibitory effects of ADP-induced platelet activation and secretion may be responsible for its antiplatelet and antithrombotic activities. Notably, the IC_50_ value of isaridin E in inhibiting ADP-induced platelet ATP release was lower than that of the inhibition of ADP-induced aggregation (Figure 3C,D), suggesting that isaridin E may specifically affect platelet secretion. However, the hypothesis that isaridin E inhibits some specific aspects of platelet secretion and activation still needs to be examined on animal models of thrombotic diseases.

Although the efficacy of platelet inhibition has been significantly improved, a major side effect of all current antiplatelet agents is still bleeding [36]. Therefore, another purpose of the presented study was to focus on finding a potent and safe new marine compound that reduces the risk of bleeding. We found that the antithrombotic effects of 50 or 100 mg/kg isaridin E are comparable to that of 10 mg/kg clopidogrel on the FeCl_3_-induced carotid artery thrombosis mouse model (Figure 5), indicating the potent antithrombotic effects of isaridin E. However, isaridin E showed no obvious influence on the blood cell counts and the blood coagulation cascade such as platelet counts, mean platelet volume, APTT, TT, PT, and FIB (Appendix A and Appendix A). Moreover, the results of tail bleeding assay showed that isaridin E significantly reduced the bleeding time compared with clopidogrel (10 mg/kg) (Figure 2). This finding is consistent with the results from measurements of coagulation parameters, suggesting that isaridin E mainly affects platelet-mediated thrombosis formation but not the coagulation cascade. Given its specific inhibitory effects on ADP-induced platelet ATP release and platelet activation, we speculate that isaridin E may reduce the risk of bleeding through specifically regulating ADP-induced platelet signaling pathways. 

It is very well recognized that PI3K/Akt pathways are crucial in platelet activation during thrombus formation [31]. Pharmacological and genetic studies have shown that the class I PI3Ks (PI3Kα, PI3Kβ, PI3Kγ, and PI3Kδ) are key signal mediators in the process of platelet activation downstream of G-protein-coupled receptors (e.g., P2Y_12_, PARs, prostaglandin receptors), GPIb-IX-V (vWF receptor), ITAM-coupled receptors (GPVI), and integrin receptors (e.g., α_2_β_1_ for collagen and α_IIb_β_3_ for fibrinogen) [37,38]. It has been demonstrated that the Ser/Thr kinase Akt is sequentially activated by specific PI3K isoforms stimulated by various membrane receptors. The PI3K/Akt signaling axis has been implicated to be upregulated in platelet hyperactivity during ischemic cardiovascular diseases [37,39]. Given their central roles in platelet activation and integrin signaling, PI3Ks have been taken as the promising target candidates for antithrombotic therapies, and some novel PI3K inhibitors have been investigated [40]. For example, the PI3Kβ inhibitor AZD6482 displayed strong anti-thrombotic effects without prolonging bleeding time or increasing blood loss in experimental arterial thrombosis models [40], indicating PI3Kβ inhibition to be a target of antiplatelet therapy verification. Here, we found that isaridin E at 50 μM significantly suppressed the increased phosphorylation of PI3K and Akt induced by ADP in Meg-01 cells without altering their total protein expressions. Based on the in vitro antiplatelet and in vivo antithrombotic effects of isaridin E, we speculated that the inhibition of PI3K/Akt pathways may be involved in the antithrombotic effects of isaridin E. However, further investigation is required to examine how isaridin E affects specific class I PI3Ks isoforms and regulates platelet activation. 

Although not specifically addressed in this study, the significant inhibitory effects of isaridin E on PI3K/Akt pathways, or on ADP-induced but not collagen- or thrombin-induced aggregation, may be related to its beneficial effects in reducing the risk of bleeding. The beneficial effects of isaridin E in reducing bleeding risks in antithrombotic therapies on disease models may provide interesting directions for future study.

## 4. Materials and Methods

### 4.1. Preparations of Isaridin E

The ascidian-derived fungal strain *Amphichorda felina* (syn. *B. felina*) sp. SYSU-MS7908 was preserved at the School of Marine Sciences, Sun Yat-Sen University, Guangzhou, Guangdong, China, and was grown on a solid rice medium (40 mL of rice, 40 mL of water with 3% artificial sea salt, and 0.3% peptone) in 200 bottles of culture flask for 28 days at room temperature [17]. The solid fermented substrate was extracted with methanol (MeOH) to obtain crude extracts, then suspended in water, and continuously extracted with ethyl acetate (EA). The EA extract (170 g) was subjected to a silica gel column and was eluted with a linear gradient of petroleum ether (PE): EA (from 9:1 to 0:10) to afford seven fractions (A–G). Fr. C was separated by column chromatography over ODS-C18 (30–40 μm, Welch Materials, Inc.) with MeOH: H_2_O from 3:7 to 10:0 to afford three subfractions (Fr.C-1~Fr.C-4). Fr. C-3 was recrystallized in ethanol to obtain compound 1 (isaridin E, 1.2 g). The purity of isaridin E were characterized by HPLC and ^1^H NMR. Its chemical structure was identified by the ^1^H and ^13^C NMR and HR-ESIMS spectra [18]. Isaridin E was dissolved in dimethyl sulfoxide (DMSO) and stored at 4 °C until use for in vitro experiments with a final concentration of DMSO of less than 0.1%. For intragastric administrations, isaridin E was dissolved in a saline solution containing 10% Tween 80 and 15% propylene glycol (used as a vehicle control) to 1.25, 2.5, 5, and 10 mg/mL, respectively. 

Isaridin E: ^1^H NMR (400 MHz, CDCl_3_) *δ* 8.14 (d, *J* = 7.5 Hz, 1H), 7.42 (d, *J* = 10.2 Hz, 1H), 7.27 (m, 2H), 7.25 (m, 2H), 7.25 (m, 1H), 5.34 (d, *J* = 10.2 Hz, 1H), 5.12 (d, *J* = 10.7 Hz, 1H), 4.65 (ddd, *J* = 10.9, 7.5, 5.0 Hz, 1H), 4.29 (d, *J* = 10.7 Hz, 1H), 4.15 (m, 1H), 4.09 (d, *J* = 7.6 Hz, 1H), 3.50 (dd, *J* = 9.8, 6.4 Hz, 2H), 3.17 (m, 1H), 3.14 (s, 3H), 3.01 (m, 1H), 2.97 (s, 3H), 2.63 (dd, *J* = 11.6, 2.8 Hz, 1H), 2.48 (m, 1H), 2.44 (m, 1H), 2.39 (m, 1H), 2.22 (m, 1H), 2.13 (m, 1H), 1.96 (m, 1H), 1.96 (m, 1H), 1.77 (m, 1H), 1.30 (m, 1H), 1.24 (m, 1H), 1.01 (d, *J* = 3.2 Hz, 3H), 0.99 (d, *J* = 3.2 Hz, 3H), 0.92 (d, *J* = 6.4 Hz, 3H), 0.89 (d, 3H), 0.87 (d, 3H), 0.87 (d, 3H); ^13^C NMR (101 MHz, CDCl_3_) *δ* 19.0, 19.6, 19.8, 20.4, 20.6, 22.1, 23.5, 24.9, 27.8, 27.8, 29.2, 29.8, 32.4, 35.2, 35.5, 35.7, 38.9, 47.3, 53.9, 57.7, 61.1, 66.6, 73.5, 127.4, 128.8, 128.9, 136.5, 168.8, 169.9, 170.1, 172.2, 173.8, 174.2. HRESIMS *m/z* 656.40183 [M + H]^+^ (calcd for C_35_H_54_O_7_N_5_, 656.40178). 

### 4.2. Animals

Adult male C57BL/6J mice (8–10 weeks old) were purchased from GemPharmatech. Animal experiments were approved by Sun Yat-sen University Animal Care and Use Committee (Approval No: SYSU-IACUC-2020-000446) and strictly followed the “Guide for the Care and Use of Laboratory Animals” issued by the Ministry of Science and Technology of China.

### 4.3. Preparation of Mouse Platelets

Washed platelets were prepared as previously described by Zhongren Ding et al. [41]. Whole blood was collected from the abdominal aorta using a syringe containing 3.8% sodium citrate (9:1, *V*/*V*) as an anticoagulant. Citrated blood (2 mL) was centrifuged at room temperature at 300× *g* for 2 min to obtain platelet-rich plasma (PRP). Platelets were obtained by the further centrifugation of PRP at 700× *g* for 2 min, and resuspended in Tyrode’s buffer (138 mM of NaCl, 2.7 mM of KCl, 1 mM of MgCl_2_, 3 mM of NaH_2_PO_4_, 5 mM of glucose, 10 mM of HEPES, pH 7.4) with 0.02 U/mL apyrase (A6410, Sigma-Aldrich, St. Louis, MO, USA) to a density of 3 × 10^8^ of platelets/mL for use.

### 4.4. Lactate Dehydrogenase (LDH) Release Assays

The possible cytotoxic effect of isaridin E on mouse platelets was assessed with the LDH Cytotoxicity Assay Kit (C0016, Beyotime Biotechnology, Haimen, China). Washed mouse platelets were pretreated with vehicle or different concentrations of isaridin E (6.5, 12.5, 25, 50, 100, 200, or 400 μM) at 37 °C for 30 min, and then collected for detection following the manufacturer’s instructions. The cytotoxic effects were determined using the percentage of LDH leakage released from the platelets (% LDH release). 

### 4.5. Platelet Aggregation and Release

The platelet aggregation and release assay were performed as previously reported [41]. Washed mouse platelets were treated with isaridin E (6.25, 12.5, 25, 50, 100, or 200 μM) or vehicle or clopidogrel (20 μM) at 37 °C for 30 min, followed by stimulation with ADP (01905, Sigma-Aldrich, St. Louis, MO, USA, 5 μM), collagen (C7661, Sigma-Aldrich, St. Louis, MO, USA, 1 μg/mL) or thrombin (T6884, Sigma-Aldrich, St. Louis, MO, USA, 0.02 U/mL). The platelet aggregation was assessed using a lumi-aggregometer (700-2, Chrono-Log, Havertown, PA, USA) with a stirring speed of 1200 rpm at 37 °C. The release of ATP was simultaneously monitored using Luciferin/luciferase reagent (395, Chrono-Log, Havertown, PA, USA). Aggregation traces and ATP release traces were monitored continuously for at least 10 min.

### 4.6. Flow Cytometric Analysis

Washed mouse platelets were pretreated with vehicle or different concentrations of isaridin E for 30 min at 37 °C, followed by stimulation with 50 μM ADP for 5 min. Platelets were then stained with FITC-conjugated anti-CD41 antibody (133904, Biolegend, San Diego, CA, USA) and PE-conjugated anti-CD62p antibody (148306, Biolegend, San Diego, CA, USA) for 20 min at 25 °C in the dark. Then, platelets were fixed in 1% paraformaldehyde solution and analyzed with a CytoFLEX flow cytometer (Beckman Coulter, Brea, CA, USA). 

### 4.7. Tail-Bleeding Assay

Mice were assigned randomly to 7 groups and intragastrically administrated with a single dose of vehicle, isaridin E (12.5, 25, 50, or 100 mg/kg), or clopidogrel (10 mg/kg) at 3 h before the cutting-tail experiment [42,43]. Tails were transected at a distal margin of 5 mm and were immediately immersed into 10 mL of saline at 37 °C. The bleeding time was recorded as when the observed bleeding ceased. Tails were cauterized if the blood flow did not cease after 1800 s; here, the bleeding time was recorded as 1800 s.

### 4.8. Hematologic and Coagulation Parameters

Mice were intragastrically administrated with a single dose of isaridin E (100 mg/kg) or vehicle at 3 h before anesthesia. Whole blood was drawn from the abdominal aorta and collected in K_2_EDTA anticoagulant tubes to evaluate the hematologic parameters with an automatic hematology analyzer (BC-2800 vet, Mindray, Shenzhen, China). The citrated blood was centrifuged at 1500× *g* (25 °C) for 15 min to obtain the platelet-poor plasma (PPP). PPP was then used to measure the coagulation parameters, including ATPP, PT, TT, and FIB, with an automated coagulation analyzer (RAC-030, Rayto, Shenzhen, China).

### 4.9. Fecl_3_-Induced Carotid Artery Thrombosis

The mouse model of carotid artery thrombosis induced by FeCl_3_ injury was performed and monitored by a stereoscopic fluorescence microscope as previously described [44]. Briefly, mice were intragastrically administrated multiple doses of vehicle or isaridin E (12.5, 25, 50, or 100 mg/kg) or clopidogrel (10 mg/kg) at 1, 24, and 48 h before the onset of FeCl_3_ injury. Mice were then anesthetized with a single dose of sodium pentobarbital (60 mg/kg, i.p., Sigma-Aldrich, St. Louis, MO, USA). Rhodamine 6G (83697, Sigma-Aldrich, St. Louis, MO, USA; 0.5 mg/mL) was injected into the right jugular vein to label platelets. The carotid artery (CA) was carefully exposed, and a black "U"-shaped plastic (3 mm) was inserted under the CA gently to block background fluorescence. Filter papers (1 × 2 mm) with 7.5% saturated FeCl_3_ solution were applied topically for 1 min, and thrombus formation in the injured CA was recorded in real time using a stereoscopic fluorescence microscope (M205FA, Leica, Wetzlar, Germany) for 12 min.

### 4.10. Western Blot Analyses

Western blotting was performed as described in our previous study [11,13]. Meg-01 cells were lysed with protein lysis buffer (P0013, Beyotime Biotechnology, Haimen, China) containing a protease inhibitor cocktail (539131, Millipore, Billerica, MA, USA) and phosphatase inhibitor cocktail (K1012, APExBIO, Houston, TX, USA). The samples (20 μg) were then separated by 8% SDS-PAGE gel and transferred onto PVDF membranes. The PDVF membranes were incubated with primary antibodies at 4 °C for 12 h. After incubating with the appropriate HRP-conjugated secondary antibodies (7076, 7074, Cell Signaling Technology, Danvers, MA, USA), blots were detected with chemiluminescent HRP substrate (P90719, Millipore, Billerica, MA, USA) using the ChemiDoc XRS+ system (BIO-RAD, Hercules, CA, USA). Primary antibodies against the following antigens were listed: PI3K (4257, Cell Signaling Technology, Danvers, MA, USA), Phospho-PI3K (ab226842, Abcam, Cambridge, UK), Akt (4691, Cell Signaling Technology, Danvers, MA, USA), Phospho-Akt (4056, Cell Signaling Technology, Danvers, MA, USA), and β-actin (66009-1, Proteintech, Rosemont, IL, USA).

### 4.11. Statistical Analysis

All data are shown as the mean ± SEM. GraphPad Prism software version 8 was utilized to perform the statistical analysis. The data were analyzed by one-way analysis of variance (ANOVA) followed by Bonferroni multiple comparison post hoc test for three or more groups and unpaired Student’s t-test for two groups. *p* < 0.05 was considered to be statistically significant.

## 5. Conclusions

In summary, the present study reveals that isaridin E attenuates thrombosis formation in an FeCl_3_-induced arteriole thrombosis model, probably through the inhibition of ADP-induced platelet activation, secretion, and aggregation, as well as influencing PI3K/Akt signaling pathways. Given that isaridin E is a safe, small molecule and exhibits a strong antithrombotic effect without increasing bleeding times, it is a promising antiplatelet and antithrombotic compound for further evaluation in the treatment of thrombotic diseases. Further studies addressing the antiplatelet and antithrombotic effects of isaridin E in disease models will be helpful to understand its pharmacological profile.

## Figures and Tables

**Figure 1 marinedrugs-20-00023-f001:**
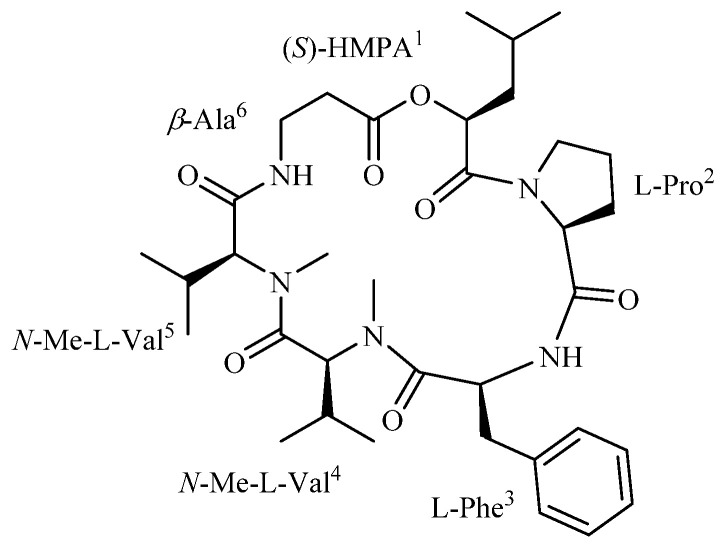
Chemical structure of compound 1 (isaridin E). HMPA (2-hydroxy-4-methylpentanoic acid).

**Figure 2 marinedrugs-20-00023-f002:**
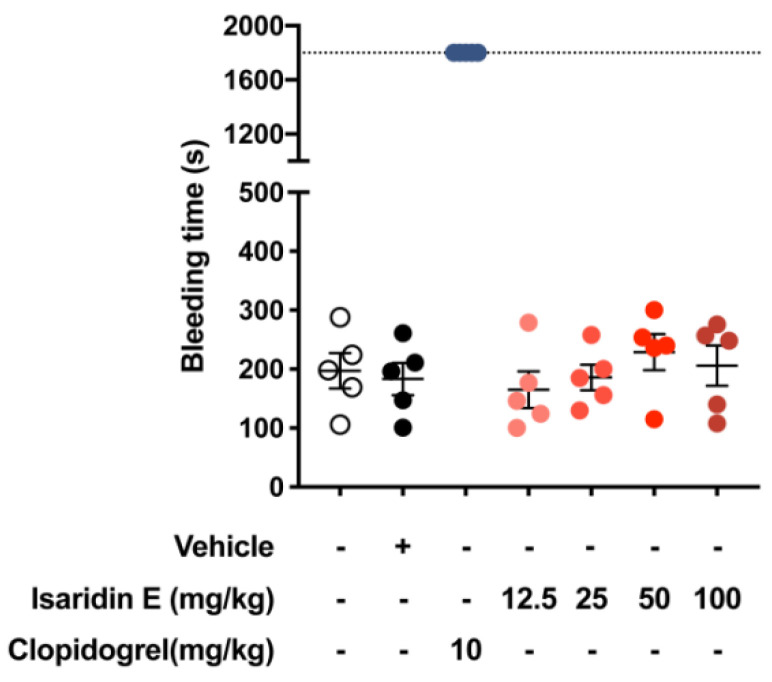
Effects of isaridin E on bleeding time. Isaridin E at doses of 12.5, 25, 50, or 100 mg/kg showed no significant effect on the bleeding time, whereas clopidogrel (10 mg/kg) significantly increased the bleeding time; *n* = 5 mice per group.

**Figure 3 marinedrugs-20-00023-f003:**
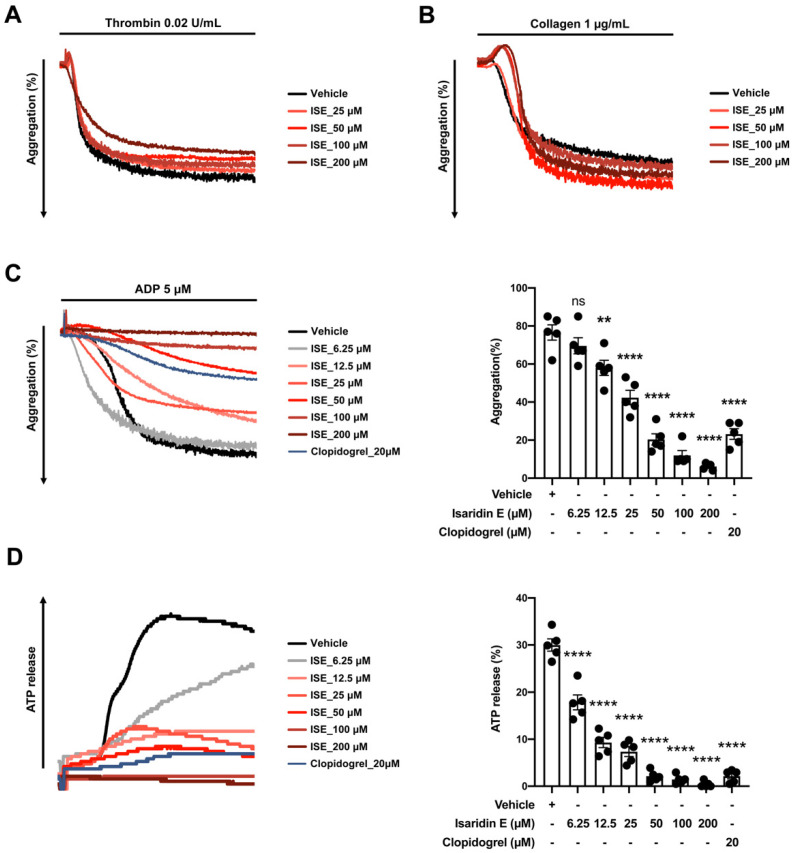
Isaridin E attenuated ADP-induced mouse platelet aggregation and secretion in vitro. Washed mouse platelets were pretreated with vehicle or isaridin E (6.25, 25, 50, 100, or 200 μM) or positive control clopidogrel (20 μM) for 30 min, followed by stimulation with thrombin, collagen or ADP. (**A**,**B**) Representative aggregation traces of platelets induced by 0.02 U/mL thrombin or 1 μg/mL collagen. (**C**,**D**) ADP (5 μM) -induced aggregation and ATP release in washed mouse platelets. ** *p* < 0.01, **** *p* < 0.0001, ns: not significant, versus vehicle control; *n* = 5 mice per group.

**Figure 4 marinedrugs-20-00023-f004:**
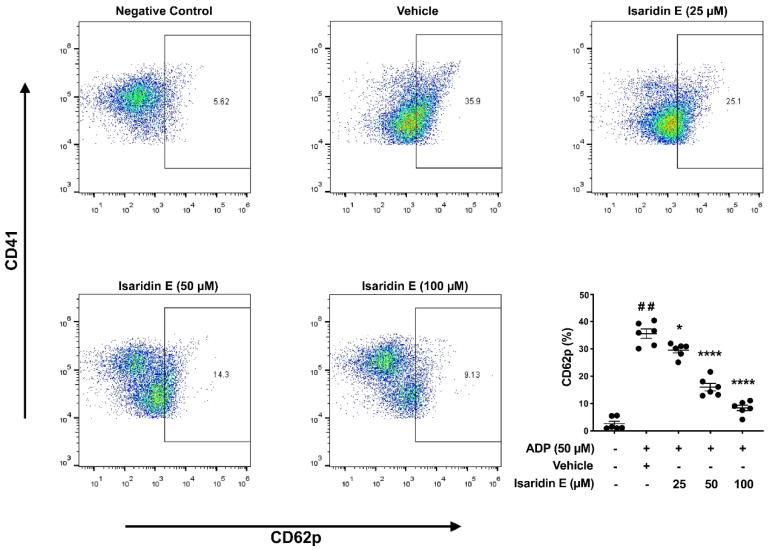
Isaridin E attenuated ADP-induced mouse platelet activation in vitro. Washed mouse platelets were pretreated with isaridin E (25, 50 or 100 μM) or vehicle at 37 °C for 30 min, followed by stimulation with 50 μM ADP for 5 min. CD62p and CD41 expression on the platelet surfaces were analyzed by flow cytometry. Untreated washed mouse platelets were served as the negative control. * *p* < 0.05, **** *p* < 0.0001, versus vehicle control; ^##^
*p* < 0.0001, versus negative control; *n* = 6 mice/group.

**Figure 5 marinedrugs-20-00023-f005:**
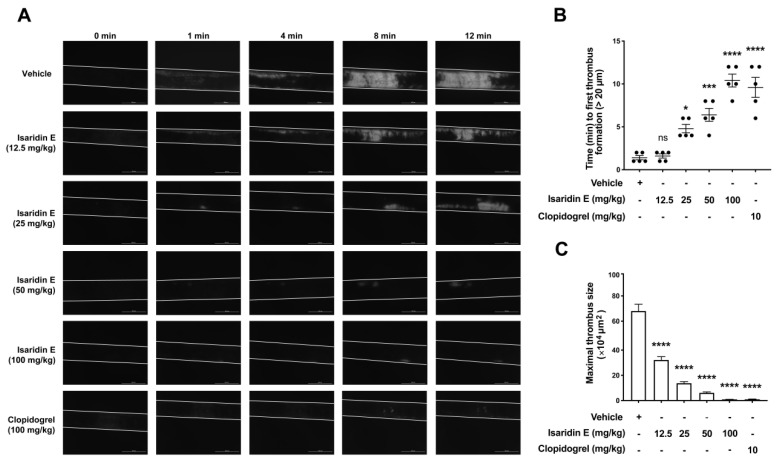
Isaridin E inhibited FeCl_3_ induced thrombus information in the carotid artery in vivo. C57BL/6J mice were administered with vehicle or isaridin E (12.5, 25, 50, or 100 mg/kg) or clopidogrel (10 mg/kg) by oral gavage at 1, 24, and 48 h before the onset of FeCl_3_ injury. A stereoscopic fluorescence microscope was used to monitor thrombus formation in the carotid artery for 12 min after FeCl_3_ injury. (**A**) Representative images of the carotid artery thrombus were captured 1, 4, 8 and 12 min after FeCl_3_ injury. (**B**) The time of forming the first thrombus (>20 μm) and (**C**) the maximal thrombus size at 12 min were recorded and analyzed. * *p* < 0.05, *** *p* < 0.001, **** *p* < 0.0001, ns: not significant, versus vehicle control. *n* = 5 independent experiments.

**Figure 6 marinedrugs-20-00023-f006:**
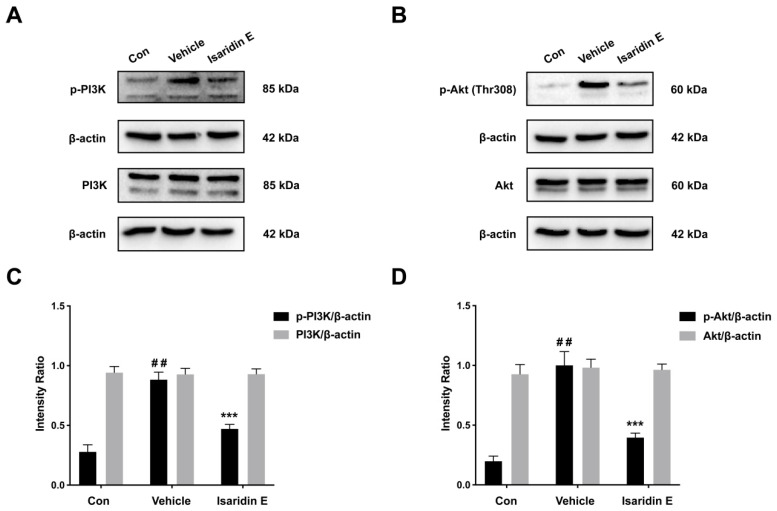
Effect of isaridin E on PI3K/Akt pathways. Meg-01 cells were pretreated with isaridin E (50 μM) or vehicle for 30 min at 37 °C, followed by stimulation with ADP (5 μM) for 5 min. The protein expressions of p-PI3K, PI3K, p-Akt, Akt, and β-actin were analyzed by Western blot (**A**–**D**). ^##^
*p* < 0.0001, versus control; *** *p* < 0.001, versus vehicle; *n* = 5.

**Table 1 marinedrugs-20-00023-t001:** Isaridin E exerted no cytotoxicity on washed mouse platelets in vitro.

Isaridin E (μM)	LDH Release (%)	*p* Value
0	0.85 ± 0.05	
6.25	0.86 ± 0.03	0.97
12.5	0.89 ± 0.04	0.54
25	0.88 ± 0.03	0.63
50	0.86 ± 0.03	0.89
100	0.88 ± 0.04	0.72
200	0.84 ± 0.02	0.83
400	0.85 ± 0.02	0.90

Note: Washed mouse circulating platelets were treated with normal saline (NS), vehicle, or isaridin E (6.25, 12.5, 25, 50, 100, 200, or 400 μM) at 37 °C for 30 min. The percentage of LDH release was determined (*n* = 6 per group).

## Data Availability

All data included in this study are available upon request by contact with the corresponding author.

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
