# Peer review of "Antiplatelet and Antithrombotic Effects of Isaridin E Isolated from the Marine-Derived Fungus via Downregulating the PI3K/Akt Signaling Pathway"

_marinedrugs, 2021, doi:10.3390/md20010023_

Round 1

Reviewer 1 Report

The peptide natural product isaridin E, of marine origin, was profiled in a number of bioassays. The results indicate that isaridin E inhibited mouse platelet aggregation and phosphorylation of PI3K and Akt in vitro, and trombus formation in vivo.
Overall, the findings are of interest and the paper should be published with minor revision: 

  1. A better drawing of isaridin E in Figure 1. In the macrocycle, the beta-Ala residue is not clearly visible.
  2. Tables- is the data sufficiently precise to be reported to two decimal places or should it be rounded up?
  3. The experimental section must include a detailed procedure for the isolation and purification of isaridin E from Beauveria felina.

Reviewer 2 Report

This manuscript is very poorly written with nearly each study included as a list of tasks that was achieved with a result that is not supported by conventional comparison to the literature. Most if not all of the studies are written up as follows...

Because of X (no explanation to the reader why X is important or reference to X), Y was done.  Isaridin E did X (no data provided and often no Figure X or Table # cited). Because of X, Isaridin E does Z.  Sadly only at a very, very few points did the authors properly describe their hypothesis, cite the literature, describe what their data indicated, and provide a scolarly comparison of their data to their controls or the field in general.

Moreover, this manuscript has not been (or if it has poorly been) proofread. It need an ENORMOUS level of editing and as is can not be accepted.

This reviewer provided yellow highlighte tracked edits.  Every sentence needs editing, these suggestions highlight issues and the authors need to carefully edit the manuscript. While one typically understands language issues, many of these edits are not due to this rather can be attributed to general sloppyness on the authors behalf.

Round 2

Reviewer 2 Report

This is a significantly improved manuscript however several of the chemical questions asked in the first review were not addressed.

Further english language editing is also needed (comments noted).

Edits tracked as notes in the attached PDF.

The authors should also read the Journals guideline and check that they match the template and format that this Journal uses. For instance, the references are not at all in the Journals format. One would expect that at least this level of editing was conducted.  Nearly every reference is different and not in the proper format.

This paper needs a line by line check. It is still full of typographical errors many are not based on the lanugage but the lack of time and effort to carefully edit the manuscript (reference provided an excellent example).
